# Oral-Health-Related Quality of Life (OHRQoL) and Anterior Open Bite in Adult Patients: A Case-Control Study

**DOI:** 10.3390/healthcare10010129

**Published:** 2022-01-09

**Authors:** Adrián Curto, Alberto Albaladejo, Alfonso Alvarado-Lorenzo

**Affiliations:** 1Pediatric Dentistry Department, Faculty of Medicine, University of Salamanca, Avenida Alfonso X el Sabio s/n, 37007 Salamanca, Spain; 2Orthodontics Department, Faculty of Medicine, University of Salamanca, Avenida Alfonso X el Sabio s/n, 37007 Salamanca, Spain; albertoalbaladejo@usal.es (A.A.); kuki@usal.es (A.A.-L.)

**Keywords:** orthodontics, anterior open bite, malocclusion, oral-health-related quality of life, oral health

## Abstract

Oral-health-related quality of life (OHRQoL) is defined as the impact of oral health on activities of daily living. Malocclusions are a public health problem with a high prevalence. Different studies have concluded that malocclusions negatively affect OHRQoL in patients of all ages. The aim of this study was to analyze the influence of having an anterior open bite on the OHRQoL of adult patients. Materials and Methods: A case-control study (1:1) was carried out with a sample size of 80 adults at the University of Salamanca in 2021. The case group (*n* = 40) was made up of patients with an anterior open bite, and the control group (*n* = 40) contained patients without an anterior open bite. OHRQoL was assessed using the Oral Health Impact Profile-14 (OHIP-14) questionnaire. The influences of gender and age on the OHRQoL of the patients were also analyzed. Results: There were no significant differences in gender or age between the case and control groups. An anterior open bite was not found to influence the OHRQoL of adult patients. Age was not shown to significantly influence OHRQoL. Female patients with an anterior open bite had higher scores in the handicap domain of the OHIP-14 questionnaire compared with male patients (*p* < 0.05). Conclusions: Anterior open bite can influence the OHRQoL of orthodontic patients. Gender can be considered an influencing factor.

## 1. Introduction

Malocclusions influence patients’ oral function and body image and cause psychological disorders [1].

Anterior open bite (AOB) is a significant malocclusion in the vertical plane and is occasionally related to other malocclusions in the sagittal plane. It is defined as the absence of vertical contact between the upper and lower incisors when the posterior teeth are closed. The prevalence of anterior open bite varies according to patient age, ethnicity, and dentition type, ranging from 1.5% to 11% [2,3,4].

Anterior open bite is one of the most complex malocclusions to treat. Its etiology is multifactorial, influenced by skeletal, dental, and oral soft tissue problems. Treatment options vary depending on the age of the patient and the causative factors [5,6].

The concept of oral-health-related quality of life (OHRQoL) describes the impact of oral conditions and the effect of dental treatment on patients. This concept is influenced by the age, gender, and health of the patient as well as their psychological state and social relationships. Conducting an OHRQoL analysis in orthodontic patients helps to determine treatment needs, giving the professional a better understanding of the patients’ expectations [7].

Different studies have evaluated the impact of orthodontic treatment on oral quality of life [1,7,8]. The impact of orthodontic treatment on OHRQoL does not follow the same pattern among patients with different types of malocclusion. When analyzing the impact of a malocclusion, it is important to consider the different domains of the patient’s oral quality [8].

Different rates and questionnaires have been used to analyze OHRQoL in orthodontic patients. In recent years, different rates of orthodontic need have been developed and validated and are now used as outcome measures. The Oral Health Impact Profile (OHIP) analyzes a person’s perception of the social impact of an oral disorder on their well-being. The OHIP questionnaire is used to analyze the influence of oral health on masticatory ability and psychosocial function. A lower score on the OHIP indicates a higher level of satisfaction with dental care [9]. The OHIP-14 questionnaire is one of the most popular methods for quantifying the impact of orthodontic treatment. The original OHIP questionnaire consists of 49 items, was developed by Slade and Spencer, and is based on Locker’s OHRQoL conceptual model [10].

Malocclusions reduce OHRQoL. The severity of a malocclusion is directly related to its impact on the patient’s oral quality of life [11]. Orthodontic treatments cause pain and discomfort for patients. The OHRQoL in orthodontics is related to the impact of dental aesthetics on patients’ social acceptance. OHRQoL is compromised during the early phases of orthodontic treatment, but it improves significantly at the end of treatment [12].

Published studies that analyzed the OHRQoL in orthodontic treatment using the OHIP-14 questionnaire observed that patient scores are lower after finishing treatment [13,14]. Previously published studies examining the influence of a malocclusion on OHRQoL in adults have concluded that there is a direct relationship between malocclusion and OHRQoL [15,16,17,18,19]. However, at present, scientific studies evaluating the impact of having an anterior open bite on OHRQoL are limited to adult patients [20,21,22].

The objective of this study was to analyze the influence of having an anterior open bite on OHRQoL in a group of adult patients and to compare the OHRQoL of these patients with a group of control patients. In addition, the influences of gender and age on patients’ OHRQoL were evaluated. The null hypothesis was that the anterior open bite does not negatively affect the OHRQoL of adult patients.

## 2. Material and Methods

### 2.1. Ethics Approval and Patient Consent

This case-control study was designed in accordance with STROBE guidelines and the Declaration of Helsinki for research involving human subjects. The research project protocol was approved by the University of Salamanca Bioethics Committee (reference number USAL_19/466).

### 2.2. Sample Size Calculation

Previously published studies assessing OHRQoL in orthodontic patients were considered when calculating the sample size [8,21,23,24,25]. Through a small pilot study of the first 12 participants (6 per group), it was observed that the variability of the different dimensions of the OHIP questionnaire is similar to that found in other studies where this same instrument was used (ranging between 0.25 and 0.50 points). An approximate value of 0.35 points was therefore determined under the assumption of homoscedasticity between groups. With a confidence level of 95% and for a power of 80%, setting a mean effect size (d Cohen = 0.60), we obtained a sample (n) of 44 subjects per group, i.e., 88 participants in total. All 88 study participants were recruited. Of the 88 study participants, six patients were lost because they did not meet the inclusion criteria. The sample size of the two groups was equalized by randomly removing two patients from the control group. Finally the sample consisted of 80 patients, 40 patients in the control group and 40 patients in the anterior open bite group.

### 2.3. Study Design

The sample consisted of 80 adult patients divided into two groups: the case group (AOB-G) (patients with an anterior open bite) and the control group (CG) (patients without an anterior open bite) (Figure 1). The patients were recruited from the Dental Clinic of the University of Salamanca. Consecutive convenience sampling was used in this study. The patients who participated in this study were selected at their first orthodontic check-up visit before starting treatment. Recruitment and selection of study participants were carried out by two orthodontic specialists with the same level of education and clinical experience. All patients were diagnosed and selected by two orthodontic specialists with more than ten years of experience.

Patient recruitment and data collection took place between January and September 2021. The patients were informed about the research project protocol, and the confidentiality of the collected data was assured. Informed written consent was obtained from the patients prior to their inclusion in the study sample. Participants who gave their signed consent were included in the study.

The inclusion criteria were as follow: adult patients (over 18 years of age) with no craniofacial anomalies, no missing teeth with the exception of third molars, a full complement of erupted teeth except for the third molars, and patients without prior orthodontic and/or dentofacial orthopedic treatment. The exclusion criteria were as follow: patients with untreated caries, patients with gingival and/or periodontal pathologies, and patients with severe dentofacial anomalies.

The age and gender of the patients were also recorded to analyze their influences on the OHRQoL. Anterior open bite was diagnosed by direct visual inspection with the teeth in centric occlusion. The diagnosis of anterior open bite was made in the absence of vertical contact between the lower and upper incisors [26].

Patients who had an anterior open bite by this definition were classified as cases for this study, and patients who were deemed negative for anterior open bite and other malocclusal traits were classified as controls.

### 2.4. Outcome Variable (Oral-Health-Related Quality of Life)

OHRQoL was evaluated using the Oral Health Impact Profile-14 (OHIP-14) questionnaire. The Spanish version of this questionnaire has been previously validated in adult patients [27]. The OHIP-14 has good reliability and precision levels [13,28].

This questionnaire was given to the patients to complete. OHIP-14 assesses the burden of oral health status on life quality across seven conceptual domains (functional limitation, physical pain, psychological discomfort, physical disability, psychological disability, social disability, and handicap) (two items per domain) of oral-health-related quality of life by asking respondents to rate the frequency of occurrence of a particular problem, as captured by the individual item. The response to each item was scored as follows: 0 = never, 1 = hardly ever, 2= occasionally, 3 = fairly often, and 4 = very often. The OHIP-14 score was calculated by summing the response codes for the 14 items. Consequently, the total OHIP-14 score ranged from 0 to 56, and domain scores ranged from 0 to 8. Higher OHIP-14 scores indicated worse and lower scores indicated better oral-health-related quality of life. Participants completed the OHIP-14 questionnaire individually.

### 2.5. Examiner Reliability Tests

The diagnosis of anterior open bite was made by two trained examiners. To analyze the reliability of the examiners, 40 patients were randomly recruited. These 40 patients were not part of this study. The patients were evaluated 3 weeks after the first examination. The intra-examiner reliability of the examiners had Kappa scores of 0.91 and 0.96. The inter-examiner reliability was adequate with a Kappa score of 0.85.

### 2.6. Statistical Analysis

Data analysis was performed using the Statistical Package for the Social Science (SPSS, version 19.0, SPSS Inc., Chicago, IL, USA). The changes observed during the study follow-up period were normally distributed, so the statistical significance of the changes was assessed with paired-samples *t*-tests. The Mann–Whitney U-test was used to analyze differences in OHRQoL between the two groups. Data are expressed as mean ± standard deviation, with a *p*-value of <0.05 indicating statistical significance.

## 3. Results

### 3.1. Characteristics of the Participants

The study sample consisted of a total of 80 patients with a mean age of 30.4 years (standard deviation ±7.02). Of the 80 patients, 42.5% (34 patients) were men and 57.5% (46 patients) were women. The patients were grouped into two study groups: the control group (CG) (*n* = 40) and the case group (AOB-G) (*n* = 40). No statistically significant differences were observed in relation to the distribution of the sample according to age and gender (Table 1).

### 3.2. Oral-Health-Related Quality of Life Analysis

We analyzed the influence of having an anterior open bite on the seven domains that make up the OHIP-14 oral quality of life questionnaire. The mean values of both groups were very similar to each other. In both study groups, there was found to be a greater impact in the psychological disability domain (AOB-G: 1.69 ± 0.60; CG: 1.45 ± 0.46). In the anterior open bite group, the domain with the second greatest impact was functional limitation (AOB-G: 1.25 ± 0.38). In comparison, in the control group, the domains with the second highest scores were physical pain and psychological discomfort (physical pain: 1.18 ± 0.29; psychological discomfort: 1.18 ± 0.27). The social disability and handicap domains were the ones with the least impact on the OHRQoL of patients in both groups (Table 2).

### 3.3. Influence of Gender

When analyzing the influence of gender on the OHRQoL, statistically significant differences were only observed in the handicap domain. It was observed that female patients described a more negative impact on their OHRQoL compared with male patients in the handicap domain (men: 0.09 ± 0.19; women: 0.24 ± 0.33) (*p* < 0.05). In the rest of the domains of the OHIP-14 questionnaire, the mean values for men and women were similar (Table 3).

### 3.4. Influence of Age

When evaluating the influence of age on OHRQoL of the patients, we decided to classify patients into three age groups: a first group of patients aged between 18 and 25 years (G1) (*n* = 23), a second group aged between 26 and 35 years (G2) (*n* = 38), and a third group aged between 36 and 50 years (G3) (*n* = 19). In this study, we did not find statistically significant differences when analyzing the influence of age on the different domains of OHRQoL of the OHIP-14 questionnaire. A slight inconclusive trend (which could be verified with larger study samples) was observed in the domains of functional limitation (lower OHRQoL in younger patients), psychological discomfort (older patients showed a more negative impact), and physical disability (OHRQoL had a greater impact on younger patients) (Table 4).

## 4. Discussion

Anterior open bite is still one of the most difficult and demanding clinical problems in adult patients. This malocclusion often results in significant esthetic and functional concerns, including difficulty incising food and speaking. Moreover, anterior open bite is accompanied by muscular and functional problems, such as incompetence of the lips and a convex facial profile [29,30]. The development of orthodontics has provided many varieties of treatment for both dental and skeletal forms of anterior open bite [30].

In recent years, the healthcare literature has placed an increased emphasis on OHRQoL [31]. Orthodontic treatment, specifically, aims to enhance oral-health-related quality of life through the correction of malocclusion, as well as improvement of dentofacial esthetics and oral function. Therefore, it is important for the orthodontic literature to evaluate patient-centered outcomes [13,32,33].

This study evaluated the impact of the anterior open bite on the oral-health-related quality of life of adult patients in the different domains of the Oral Health Impact Profile-14 (OHIP-14) questionnaire. The OHIP-14 questionnaire has been validated for adult patients and has been widely used in other studies to assess oral quality of life in orthodontic patients [13,34]. The limitation of the OHIP-14 questionnaire is that it does not assess the reasons for the impacts recorded in the OHRQoL. These causes could be related to different oral health conditions [14,35]. Participants in this study were selected if they did not have caries, gingival or periodontal pathology, severe dentofacial anomalies, or a malocclusion other than anterior open bite, suggesting that the results obtained are not confounded with other oral health problems.

The malocclusions that have the greatest impact on the OHRQoL of patients are anterior crossbite, dental crowding, and Class III malocclusion [36]. In the scientific literature, there are few articles that analyze the impact of having an anterior open bite on OHRQoL in adult patients, with most evaluating child and/or adolescent patients [15,16,17,20,21,22]. This can be explained, in part, by the fact that, today, the majority of orthodontic patients are children; however, increasingly more adult patients are requesting orthodontic treatments. In 2019, Pithon MM et al. conducted a single-blinded, randomized, controlled trial aimed at evaluating OHRQoL in children before, during, and after treatment of anterior open bite, compared to untreated child patients. It analyzed a total of 80 child patients. It concluded that treatment of anterior open bite had a positive impact on the OHRQoL of the patients, with statistically significant differences observed [21]. Similar results were observed by Ramos-Jorge J et al., in 2015 [22]. Numerous factors have been suggested to influence patient satisfaction with orthodontic treatment. A systematic review by Pachêco-Pereira C et al., in 2015, concluded that patient satisfaction after completion of orthodontic treatment was associated with the aesthetic results achieved and with the perceived psychological benefits of treatment [37].

The novelty of this study was the evaluation of the oral quality of life of adult patients with anterior open bite and the comparison of these results with those of a control group.

We conclude that for patients with anterior open bite, the worst impacts on their oral quality of life are in the psychological disability domain of the OHIP-14 questionnaire. These results are consistent with those presented by Chen M et al., in 2015. Chen et al. found that patients with malocclusions experienced greater negative impacts in the psychological disability and psychological discomfort domains of the OHIP-14 questionnaire. The author concluded that orthodontic treatment significantly improves OHRQoL in adult patients [38].

Other studies have evaluated the impact of other malocclusions on the oral quality of life of patients [15,16,17,18,19,20,21,22,35,38]. Masood M et al. in 2014 evaluated the impact of having a posterior crossbite on OHRQoL in young patients (143 patients aged 15–25 years). This study, like ours, also used the OHIP-14 questionnaire. They concluded that having a posterior crossbite significantly influences the oral quality of life of patients. They found statistically significant differences in all domains of the OHIP-14 questionnaire when comparing patients with a posterior crossbite and control patients. In this study, the authors observed that the OHRQoL dimensions of the OHIP-14 questionnaire with the highest scores in patients with crossbite were psychological discomfort (4.24 ± 1.69), functional limitation (3.89 ± 1.95), and physical pain (3.45 ± 1.56). In agreement with our study, Massod M et al. observed that the social disability dimension of the OHIP-14 questionnaire was the dimension with the lowest impact (2.85 ± 2.06), as we observed in our study (0.15 ± 0.28). Patients with a posterior crossbite had higher scores in all dimensions and in the total score of the OHIP-14 questionnaire, showing a more negative impact on their OHRQoL. It is necessary to emphasize that this author evaluated adolescent and young adult patients; in our study, we have evaluated patients over 18 years of age [35].

When analyzing the influence of gender on OHRQoL in this study, statistically significant differences were only observed in the handicap domain of the OHIP-14 questionnaire. In this domain, female patients experienced a more negative impact on their OHRQoL compared with male patients. These results are consistent with those published by Silvola AS et al. in 2020. These authors evaluated the influence of gender on oral quality of life in different types of malocclusions (including the anterior open bite) using the OHIP-14 questionnaire. In this case, they observed that women with an anterior open bite experienced a more negative impact on their oral quality of life compared with men [16].

When evaluating other malocclusions, such as those in skeletal class III, different published studies concluded that there are no significant differences based on gender in OHRQoL [39,40]. Rezaei F et al., in 2019, analyzed the OHRQoL of adult patients diagnosed with skeletal class III before and after undergoing orthognathic surgery. This study used, like this study, the OHIP-14 questionnaire. This study analyzed OHRQoL at three time points: before orthodontic treatment, before orthognathic surgery (during orthodontic treatment), and after orthognathic surgery. The mean score of the OHIP-14 questionnaire in males was higher than in females “before orthodontic treatment” and “after orthognathic surgery”; however, the mean score in females was higher than in males “before orthognathic surgery”. However, the difference in this regard was not significant between males and females in any group (*p* > 0.05) [39].

Patients with an anterior open bite are more limited in their oral functions. The need of patients to increase their oral quality of life may be related to their need for orthodontic treatment for their malocclusion. Adult orthodontic patients who have received treatment to correct anterior open bite describe satisfaction with their treatment [41]. Analysis of the influence of having an anterior open bite on OHRQoL provides further insight into the need for orthodontic treatment.

In this study, age was not observed to influence OHRQoL in adult patients with anterior open bite; however, Masood M et al. observed a negative association between age and impact on OHRQoL in adolescent and young adult patients with crossbite, with the impact of crossbite decreasing with increasing age [35]. Our results are consistent with those reported by other scientific studies [42].

In this study, it was expected that the impact of anterior open bite on oral quality of life would be significantly higher in female patients compared with male patients, based on previously published studies [16,25]. In our study, when analyzing the influence of gender on the OHRQoL, statistically significant differences (*p* < 0.05) were only observed in the handicap domain (Men: 0.09 ± 0.19; Women; 0.24 ± 0.33).

Silvola AS et al. evaluated, using the OHIP-14 questionnaire, the influence of gender on the impact on OHRQoL in adult patients with different malocclusions. In the case of anterior open bite, this study concluded that there were statistically significant differences (*p* < 0.01) in the dimension of functional limitation. In this case, it was the female patients who described a worse impact on the functional limitation dimension compared to the male patients. In the other dimensions of the OHIP-14 questionnaire, no statistically significant differences were observed. In our study, we observed that female patients described a more negative impact on their OHRQoL compared with male patients in the handicap domain (*p* < 0.05). This study by Silvola AS only analyzed anterior open bite in 13 female patients and 8 male adult patients [16].

Clinical relevance of this study: Dental professionals need to understand the impact that malocclusions (in this case the anterior open bite) can have on their patients’ OHRQoL. Specialists should be knowledgeable about the OHRQoL of adult patients to ensure that treatment also focuses on the psychological aspects of the patients. Information about patients’ OHRQoL will help us make decisions about their dental treatment.

The conclusions of this research project should be considered alongside the limitations described below. One of this study’s limitations was that we did not analyze the influence of anterior open bite treatment on the OHRQoL of patients, and we did not evaluate OHRQoL at different points in time during orthodontic treatment. Another limitation of this study is that it did not assess the influence of different sociodemographic factors (e.g., level of education or employment status) on the OHRQoL of adult patients with anterior open bite. We have also not analyzed the influence of the etiology of anterior open bite on the OHRQoL of adult patients.

It is also necessary to analyze the impacts of other malocclusions on the OHRQoL of patients and to expand the study sample and the age range (also analyzing child and adolescent patients).

## 5. Conclusions

When analyzing the influence of having an anterior open bite on the OHRQoL of adult patients, no statistically significant differences were observed compared with control patients. Female patients were shown to experience a more negative impact on their OHRQoL in the handicap domain of the OHIP-14 questionnaire. Age was not found to significantly influence OHRQoL.

## Figures and Tables

**Figure 1 healthcare-10-00129-f001:**
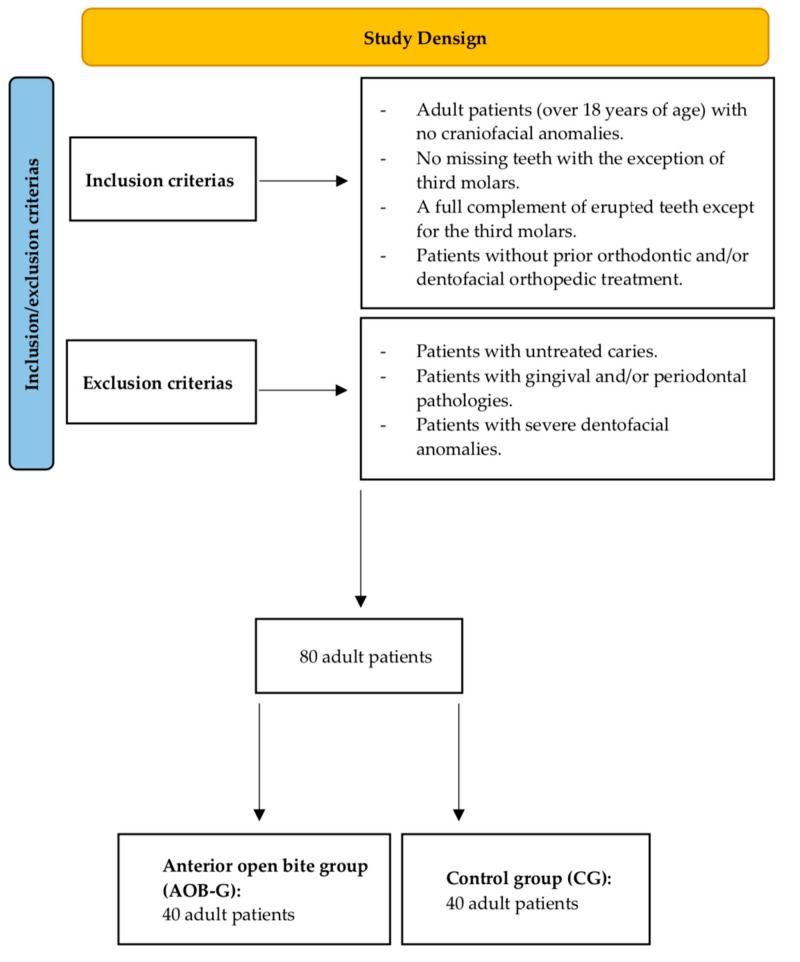
Flow chart with the study design.

**Table 1 healthcare-10-00129-t001:** Sociodemographic characteristics of anterior open bite patients and controls (*n* = 80).

		Descriptors			Statistical Hypothesis Test	
		Total Sample (*n* = 80)	Anterior Open Bite Group (AOB-G) (*n* = 40)	Control Group (CG) (*n* = 40)	Value	*p*-Value
**Gender**	**Female**	57.5% (46)	60.0% (24)	55.0% (22)	Chi2 = 0.20 ^NS^	0.651
	**Male**	42.5% (34)	40.0% (16)	45.0% (18)		
**Age (years)**	**Mean (SD)**	30.4 (±7.02)	30.6 (±7.61)	30.3 (±6.47)	T = 0.17 ^NS^	0.862

NS = Not significant at 5% (*p* > 0.05).

**Table 2 healthcare-10-00129-t002:** Differences in the domains of the OHIP-14 questionnaire by group (*n* = 80).

Domains	Mean (Standard Deviation)		Statistical Hypothesis Test		Effect Size: R^2^
	Anterior Open Bite Group (AOB-G) (*n* = 40)	Control Group (CG) (*n* = 40)	Value/Z_U_/	*p*-Value	
**Functional limitation**	1.25 (0.38)	1.16 (0.31)	1.17 ^NS^	0.242	0.016
**Physical pain**	1.18 (0.24)	1.18 (0.29)	0.31 ^NS^	0.759	0.000
**Psychological discomfort**	1.20 (0.35)	1.18 (0.27)	0.16 ^NS^	0.876	0.002
**Physical disability**	0.34 (0.46)	0.26 (0.34)	0.41 ^NS^	0.680	0.009
**Psychological disability**	1.69 (0.60)	1.45 (0.46)	1.83 ^NS^	0.067	0.048
**Social disability**	0.15 (0.28)	0.16 (0.31)	0.06 ^NS^	0.949	0.000
**Handicap**	0.15 (0.28)	0.20 (0.30)	0.91 ^NS^	0.361	0.008
**Total OHIP**	5.96 (0.54)	5.59 (0.32)	1.05 ^NS^	0.250	0.014

NS = Not significant (*p* > 0.10).

**Table 3 healthcare-10-00129-t003:** Differences in the domains of the OHIP-14 questionnaire by gender (*n* = 80).

Domains	Mean (Standard Deviation)		Statistical Hypothesis Test		Effect Size: R^2^
	Men (*n* = 34)	Women (*n* = 46)	Value /Z_U_ /	*p*-Value	
**Functional limitation**	1.21 (0.37)	1.21 (0.33)	0.24 ^NS^	0.806	0.000
**Physical Pain**	1.16 (0.27)	1.18 (0.27)	0.46 ^NS^	0.642	0.002
**Psychological discomfort**	1.24 (0.35)	1.15 (0.28)	1.02 ^NS^	0.309	0.018
**Physical disability**	0.28 (0.39)	0.32 (0.41)	0.35 ^NS^	0.726	0.002
**Psychological disability**	1.59 (0.53)	1.55 (0.56)	0.40 ^NS^	0.690	0.001
**Social disability**	0.10 (0.21)	0.20 (0.34)	1.00 ^NS^	0.318	0.025
**Handicap**	0.09 (0.19)	0.24 (0.33)	2.19 *	0.029	0.068
**Total OHIP**	5.67 (0.17)	5.85 (0.25)	1.16 ^NS^	0.262	0.010

NS = Not significant (*p* > 0.05); * = Significant (*p* < 0.05).

**Table 4 healthcare-10-00129-t004:** Differences in the domains of the OHIP-14 questionnaire as a function of age (*n* = 80).

Domains	Mean (Standard Deviation)			Statistical Hypothesis Test		Effect Size: R^2^
	18–25 Years (G1) (*n* = 23)	26–35 Years (G2) (*n* = 38)	36–50 Years (G3) (*n* = 19)	Value/Z_U_/	*p*-Value	
**Functional limitation**	1.20 (0.29)	1.26 (0.40)	1.11 (0.27)	2.74 ^NS^	0.254	0.034
**Physical Pain**	1.17 (0.29)	1.17 (0.24)	1.18 (0.30)	0.03 ^NS^	0.984	0.000
**Psychological discomfort**	1.15 (0.28)	1.16 (0.29)	1.29 (0.38)	2.14 ^NS^	0.342	0.034
**Physical disability**	0.41 (0.47)	0.29 (0.40)	0.18 (0.30)	2.78 ^NS^	0.249	0.043
**Psychological disability**	1.50 (0.34)	1.62 (0.62)	1.55 (0.60)	0.20 ^NS^	0.904	0.009
**Social disability**	0.15 (0.32)	0.14 (0.31)	0.18 (0.25)	1.18 ^NS^	0.554	0.003
**Handicap**	0.20 (0.33)	0.18 (0.29)	0.13 (0.23)	0.29 ^NS^	0.866	0.007
**Total OHIP**	5.78 (0.24)	5.82 (0.38)	5.62 (0.21)	1.15 ^NS^	0.624	0.002

NS = Not significant (*p* > 0.05).

## Data Availability

The data presented in this study are available on request from the corresponding author.

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
