# Peer review of "Oral-Health-Related Quality of Life (OHRQoL) and Anterior Open Bite in Adult Patients: A Case-Control Study"

_healthcare, 2022, doi:10.3390/healthcare10010129_

Round 1
Reviewer 1 Report
The authors of the article “Oral-Health-Related Quality of Life (OHRQoL) and Anterior Open Bite in Adult Patients: A Case Control Study“ investigated the correlation of open bite with regard to OHRQoL. For this purpose, a case group with 40 probands and control group with also 40 probands and no anterior bite were examined. Introduction is well written and provides an overview with outlining significance and aim of the study. Hypothesis is clearly outlined. Please outline in detail how and by whom patients were recruited. You talked about 2 examiners. But who and how many else recruited probands at first orthodontic check-up visit? Were the examiners orthodontists at same education level? The authors should consider a flowchart diagram regarding study design with outlining inclusion/exclusion criterias. On the one hand, age was not shown to significantly influence OHRQoL. On the other hand you concluded that age can be considered influencing factor. Is it a conclusion for your study or compared to other studies? Please describe more clearly. English language does not need corrections.
Reviewer 2 Report
First of all, thank you for the opportunity to review this case study. The aim of the following study was to explore the influence of having an anterior open bite on the OHRQoL of adult patients.
The work is flowing nicely however there is a lot of ambiguity. The idea is good, but the realization is insufficient.
The following are suggestions for the present manuscript:
Materials and methods:
- Please show a sample size calculation.
- It is normal to expect open bite have an impact on quality of life. This has been confirmed many times by various studies. The authors only diagnosed anterior open bite, but whether they divided open bite according their etiology or some other classification. Please consider this to improve the quality of work.
- Were any other demographic data taken (education, income, employment status)?
- Author write that they used “…One-way analysis of variance (ANOVA) with the Mann-Whitney U-test was used to analyse differences in OHRQoL between the two groups.”
- Mann–Whitney U test is a nonparametric test? This test cannot be used with the ANOVA test which is parametric?
Results:
- Why are there no results for total OHIP-14 in Tables 2, 3 and 4?
Discussion:
- The discussion is short and vague. Please discuss the results more clearly with similar studies.
- “Other studies have evaluated the impact of other malocclusions on the oral quality of life of patients.” What other studies, please refer to them.
- Indicate the strength of the study, if any.
Round 2
Reviewer 1 Report
The reviewer would like to thank the authors for their revision.
Reviewer 2 Report
I still don’t understand how the sample size was calculated.
The initial part of the discussion is more about the introduction than the discussion. You need to explain your results.
The authors did not indicate the strength of the study.
